# Role of Alexithymia in Predicting Internet Novel Addiction through Boredom Proneness

**DOI:** 10.3390/ijerph19148708

**Published:** 2022-07-17

**Authors:** Yuying Liu, Lei Chen, Zhiyan Wang, Ge Guo, Mingming Zhang, Shunsen Chen

**Affiliations:** 1Fujian Key Laboratory of “Applied Cognition & Personality”, School of Educational Science, Minnan Normal University, Zhangzhou 363000, China; ujunglyy@gmail.com; 2Research Center of Brain and Cognitive Neuroscience, Liaoning Normal University, Dalian 116029, China; chenleicclccl@outlook.com (L.C.); wangzylnnu@gmail.com (Z.W.); 3College of Early Childhood Education, Yango University, Fuzhou 350015, China; gguo@ygu.edu.cn

**Keywords:** college student, alexithymia, internet novel addiction, boredom proneness

## Abstract

With the development of the internet, people’s pursuit of reading entertainment has enriched internet novels, but the relevant influencing factors are still unclear. Therefore, we recruited 344 Chinese college students and employed a questionnaire survey to explore the relationship between alexithymia, boredom proneness, and internet novel addiction. The results showed that (1) there was no significant difference between female and male college students in terms of alexithymia and boredom proneness, whereas male college students had a higher total score of internet novel addiction than females. (2) There were significant positive correlations between alexithymia, boredom proneness, and internet novel addiction. (3) Boredom proneness played a partial mediating role in the impact of alexithymia on the internet novel addiction. Taken together, alexithymia may directly and indirectly predict internet novel addiction through boredom proneness.

## 1. Introduction

The number of internet users in China reached 1.032 billion, and the internet penetration rate reached 73% as of December 2021 [1]. With the rapid development of the internet and e-book culture, many college students read internet novels as a leisure activity. As college students can read novels anytime and anywhere, such behavioral trends are easy to make some with weak self-control immersed in the world of the internet. Internet novel addiction refers to the immersion in internet novels for a long time [2]. In the era of internet literature, novels are a popular pastime among college students. As internet novels have concealment, the act of reading these novels costs less than playing internet games, and the explicit behavior and range of activities are more hidden. The behaviors of internet novel addiction are sometimes not easily discovered because of its concealment, let alone being valued by society. Studies in China have also shown that personality traits, narrative transportation, and flow experience are the main causes of internet novel addiction [3]. 

Alexithymia was first described and named by Sifneos in 1973. The salient features of the construct are: (1) difficulty identifying and describing feelings; (2) difficulty differentiating between emotional states and physical sensations; (3) constricted imaginative activity, as evidenced by a paucity of fantasies and dreams; and (4) an externally oriented cognitive style [4]. In the current research stage, alexithymia is generally regarded as a lack of emotional perception, processing, and regulation, which is a state response. Generally, alexithymia is not a symptom of mental illness, but a relatively stable personality trait. More researchers tend to study the impact of alexithymia as a personality trait on individuals’ physical and mental health, social behavior, addictive behavior, and interpersonal relationships. For example, in the context of COVID-19, studies have shown that compassion disorder is associated with physical and mental health (e.g., somatization, anxiety, stress, post-traumatic stress disorder, and depression) [5,6,7,8]. Additionally, Estévez et al. (2021) showed that alexithymia and attachment style were strongly associated with addictive behaviors [9]. There was also a positive correlation between alexithymia and smartphone addiction [10,11,12,13,14,15]. Furthermore, interpersonal relationships have been shown to have an impact on partner relationships [16]—alexithymia has been shown to predict symptoms of internet addiction [17,18,19]. 

Many college students are more likely to feel “bored” on days without courses, thinking that they have nothing to do, feeling idle and empty, not being interested in learning [20]. Swinkles (1995) has divided boredom into state boredom and trait boredom [21]. State boredom refers to when an individual in a certain situation produces a comparatively short-lived boring emotional experience at a certain moment; trait boredom, is a state of mind, often referred to as a proneness to be bored with stability and individual differences, primarily motivated by individual’s intrinsic motivation (e.g., meaningless life) causes, imperfect cognitive processes, or inefficiency in attention as important factors in producing “idiosyncratic boredom” [22]. In this study, the boredom proneness refers to stable individual differences in boredom emotional responses and behaviors among relatively persistent personality traits. The relationship between boredom proneness and hedonistic and impulsive behaviors has been studied, such as unhealthy diets [23], aggression [24,25], alcoholism [26,27], substance abuse [28], unsafe driving [29], problem gambling [30,31], and sexuality [32,33]. Furthermore, studies have shown that difficulty identifying emotions and describing affective difficulties in alexithymia have a significant predictive effect on individual differences in boredom proneness [34,35], which is also significantly and positively related to internet addiction [36,37]. 

Moreover, Eastwood et al. (2007) suggests that the lack of emotional awareness is the main reason for the experience of boredom [34]. Therefore, alexithymia may be one of the factors affecting the experience of boredom that is due to the inability to better experience and express their emotions [34]. Extant literature has found that alexithymia has a significant predictive effect on boredom proneness, alexithymia and internet addiction are interrelated, and boredom proneness and internet addiction are significantly positively correlated. However, whether internet novel addiction as a subcategory of internet addiction is related to alexithymia and boredom proneness requires further study. Therefore, to explore the relationship between college students’ alexithymia, internet novel addiction, and boredom proneness, 344 college students were recruited for the present study. We hypothesized that boredom proneness may play a mediating role in the prediction of alexithymia to internet novel addiction.

## 2. Materials and Methods

### 2.1. Participants

The determination of sample size was conducted using WebPower [38]. With the path coefficients of 0.3, power of 0.95, and 3 variables, the smallest sample size required was 290. A total of 354 college students participated in this study, surveyed by the WJX (https://www.wjx.cn/) using simple sampling. All 354 questionnaires were returned, 344 of which were valid, with an effective rate of 97.18%. There were 169 males (49.13%) and 175 females (50.87%): 67 first-year students (19.48%), 80 second-year students (23.26%), 88 third-year students (25.58%), and 109 fourth-year students (31.69%). The male–female ratio in all grades was approximately in line with the real situation of the university surveyed. They signed the written informed consent before participating in this survey, and the study was approved by the Ethics Committee of Minnan Normal University. 

### 2.2. Toronto Alexithymia Scale (TAS-20)

The Chinese version of TAS-20 introduced by Yuan et al. (2003) [39], which has 20 items, scored on a scale of 1 to 5 (1 = strongly disagree, 5 = strongly agree). In the TAS-20, items 4, 5, 10, 18, and 19 were scored in reverse. The scale includes three factors. Factor 1 (i.e., difficulty identifying feelings) comprises items 1, 3, 6, 7, 9, 13, and 14. Factor 2 (i.e., difficulty describing feelings) contains items 2, 4, 11, 12, and 17. Factor 3 (i.e., externally oriented thinking) has items 5, 8, 10, 15, 16, 18, 19, and 20. The TAS-20 score was obtained by summing the responses to all items. The reliability coefficient α of the TAS-20 for this study was 0.676.

### 2.3. Undergraduates’ Internet Novel Addiction Questionnaire

We used the “Undergraduates’ Internet Novel Addiction Questionnaire” compiled by Song et al. (2013) [39], which contains five dimensions and a total of 23 items, scored on a scale of 1 to 5 (1 = strongly disagree, 5 = strongly agree). The scale includes five factors—Factor 1 (i.e., health and learning status): items 1, 9, 10, 16, 20, and 23; Factor 2 (i.e., interpersonal relationships): items 2, 8, 11, and 17; Factor 3 (i.e., tolerance): items 3, 7, 12, 15, 18, and 21; Factor 4 (i.e., time management): items 4, 6, 13, and 22; and Factor 5 (i.e., withdrawal response): items 5, 14, and 19. The higher the degree of internet novel addiction, the higher the score [40]. The reliability coefficient α of this internet addiction questionnaire for the current study was 0.937.

### 2.4. Boredom Proneness Scale for College Students (BPS)

We used the “Questionnaire on Boredom Proneness of College Students” revised and compiled by Huang et al. (2010) [22], which is composed of 30 items, scored on a scale of 1 to 7 (1 = strongly disagree, 4 = neutral, 7 = completely agreed). It includes two dimensions: external stimulation and internal stimulation. In the BPS, items 17, 18, 19, 20, 21, 22, 23, and 24 were scored in reverse. The higher the score, the stronger the proneness to boredom, of which the external stimulus dimension reflects the individual’s low perception of environmental stimuli, including four factors: monotony, constraint, loneliness, and tension. The internal stimulus dimension reflects the individual’s ability to create interest activities by themselves, including two factors: self-control and creativity. The reliability coefficient α of the BPS for this study was 0.759.

### 2.5. Statistical Analyses

SPSS 23.0 (IBM, Armonk, NY, USA) was used for reliability analysis, common method bias test, independent sample *t*-test, and Pearson correlation analysis. The SPSS plugin PROCESS was used for mediation effect analysis. 

## 3. Results

### 3.1. Common Method Bias Test

Harman’s single factor test for all items was conducted [41]. The unrotated result showed that 14 factors had an eigenvalue higher than 1. The explained variance of the first factor was 30.41%, which was lower than the critical value of 40%, indicating that the data in this study were less affected by the common method bias.

### 3.2. Sex Differences in Alexithymia, Boredom Proneness, and Internet Novel Addiction

An independent sample *t*-test was conducted on the sex differences in internet novel addiction, boredom proneness, and alexithymia (see Table 1).

There was no significant difference between females and males in the overall score of alexithymia (*t* = 0.38, *p* > 0.05). Males and females had a significant difference in the overall score of internet novel addiction (*t* = 2.30, *p* < 0.05), with male internet novel addiction (M = 69.43, SD = 18.30) slightly higher than female (M = 65.07, SD = 16.82). On the overall score for boredom proneness, there was no significant difference between males and females (*t* = 0.46, *p* > 0.05). 

The table also shows no significant differences between males and females in three dimensions of alexithymia: difficulty recognizing emotions (*t* = 0.04, *p* > 0.05), difficulty describing emotions (*t* = 0.94, *p* > 0.05), and extroverted thinking (*t* = 0.07, *p* > 0.05). There were significant sex differences in the three dimensions of internet novel addiction: health and learning (*t* = 2.58, *p* < 0.05), interpersonal relationships (*t* = 2.68, *p* < 0.05), and time management (*t* = 4.63, *p* < 0.05). However, there was no significant sex differences in tolerance (*t* = -0.53, *p* > 0.05) and withdrawal responses (*t* = 1.71, *p* > 0.05). The two dimensions of boredom propensity, internal (*t* = 0.30, *p* > 0.05) and external stimuli (*t* = 0.26, *p* > 0.05), did not significantly differ between female and male.

### 3.3. Correlations between Alexithymia, Boredom Proneness, and Internet Novel Addiction

In this study, the relationships between college students’ alexithymia, internet novel addiction, and boredom proneness were analyzed by the Pearson correlation method (see Table 2).

The total score of alexithymia and of internet novel addiction showed a significant positive correlation (*r* = 0.61, *p* < 0.001). The total score of boredom proneness a significant positive correlation with the total score of alexithymia (*r* = 0.63, *p* < 0.001), which was significantly positively correlated with the total score of internet novel addiction (*r* = 0. 53, *p* < 0.001).

### 3.4. Mediating Role of Boredom Proneness

According to Hayes’ mediation effects test and analysis program [42], a bootstrapping of 5000 times was performed with a 95% confidence interval (see Table 3 and Figure 1).

The results showed that a × b was significant (a = 0.63, b = 0.23, a × b = 0.14, *p* < 0.001); in the analysis procedure that indicated the existence of a mediating path, the boredom proneness of mediating effect was significant. Moreover, after controlling the mediating variable boredom proneness, a significant direct effect of alexithymia on the internet novel addiction was found (c’ = 0.46, *p* < 0.001). The total effect was also significantly positive (c = 0.61, *p* < 0.001). Therefore, boredom disposition played a partial mediating role in the effect of alexithymia on the internet novel addiction.

## 4. Discussion

### 4.1. Sex Differences in Alexithymia, Boredom Proneness, and Internet Novel Addiction

The results of this study showed that there was significant difference between males and females in terms of internet novel addiction. Male college students were more addicted to internet novel than their female counterparts. A study shows that male college students are less focused on health and time management than female college students [43], which might be the internal reasons for male college students to be more addicted to internet novel. Note that Zhang et al. (2017) did not find the significant correlation between sex and internet novel addiction [3]. Hence, sex difference should also be considered in the future interventions or preventive strategies to internet novel addiction. 

There was no significant difference between male and female college students in terms of alexithymia. Studies have shown that college students of different genders have no significant differences in the overall scores of emotional discrimination, emotional description, extroverted thinking, and alexithymia [44]. The results of this study are no different from those of previous studies [45,46,47], and further demonstrate that there are no significant differences in sex in alexithymia. 

In terms of boredom proneness, there were no significant sex differences, but previous studies have shown that female college students score lower overall for boredom proneness than male counterparts [48]. This study contradicts the conclusions of the previous study [49,50]. It is speculated that the pandemic has isolated college students, and the difference in boredom between males and females has decreased, resulting in insignificant test results. 

### 4.2. Correlations between Alexithymia, Boredom Proneness, and Internet Novel Addiction

In the present study, the total score of college students’ alexithymia and of internet novel addiction showed a significant positive correlation. This indicates that the higher the degree of alexithymia, the more obvious the addiction to internet novels will be. This is becoming increasingly common for college students who feel distanced from their classmates or roommates because they cannot express their emotions well in college life. Owing to setbacks in interpersonal relationships, they are addicted to the internet world.

In addition, the total score of alexithymia and of boredom proneness in this study showed a positive correlation, which is consistent with the previous studies’ results. Moreover, previous studies have found that boredom proneness is positively correlated with alexithymia. For example, Li investigated the relationship between alexithymia with gaming disorder, depression, boredom, and loneliness in university students [51]. They found a significantly positive correlation between alexithymia and boredom. Craparo et al. (2020) reported that a significant relationship between boredom and alexithymia [52]. It is suggested that the lack of ability to describe and recognize emotions and to distinguish between emotions and somatic feelings is prone to boredom.

We also found that the total score of boredom proneness showed a positive and significant relationship with the total score of internet novel addiction. This indicates that the higher the proneness to boredom college students, the more obvious internet novel addiction behavior. This shows that internet novel addiction belongs to a subcategory of internet addiction and, thus, boredom proneness has a clear predictive effect on the internet addiction [36]. Some of these conclusions can be applied to the relationship between boredom proneness and internet novel addiction.

On the other hand, Vodanovich and Kass (1990) extracted five orthogonal factors from the BPS, including external stimulation, internal stimulation, constraints, affective response, and perception of time [53]. In the present study, the BPS only have two dimensions: external stimulation and internal stimulation. Huang et al. (2010) attributes time perception to self-control in internal stimulation; that is, high scorers tend to lose their attention and it is difficult to maintain long-term attention to a single person, object, or thing [22]. Therefore, according to the attention theory of boredom proneness [54], the root of an individual’ s susceptibility to boredom lies in the interruption of attention regulation, that is, the inability to maintain and regulate attention [35,55]. The present results also fit the person–affect–cognition–execution model [56], in which boredom proneness affects individuals’ cognitive and emotional responses, and enhances their desire to reduce their boredom experience by using a certain application. Specifically, individuals with high levels of boredom proneness have difficulty directing their attention to a stimulus or trying other things to alleviate boredom, and they tend to immerse themselves in internet novels to escape from real problems and boredom, and therefore tend to develop internet novel addiction.

### 4.3. Mediating Role of Boredom Proneness between Alexithymia and Internet Novel Addiction

Most importantly, in the study, boredom proneness was shown to play a mediating role in the impact of alexithymia on internet novel addiction. The higher the degree of boredom proneness and the degree of alexithymia, the more likely individuals are to develop internet novel addiction. Alexithymia makes the behavior of internet novel addiction more frequent through the mediating role of boredom proneness. This result is similar to previous findings that alexithymia is positively related to boredom proneness, and problematic behaviors are positively correlated with boredom proneness [57,58,59,60]. This result can be explained as follows. Some researchers suggest that boredom can be seen as a psychological conflict whose core feature lies in the lack of emotional awareness, and that people’s inability to directly and deeply experience his or her emotions may be the cause of trait boredom [61]. Individuals with high level of alexithymia are not able to identify, describe, and monitor well their own emotions and those of others, resulting in their internal needs and desires being suppressed [62]. As a result, it is difficult for them to find purpose and meaning in their lives and to gain inspiration and motivation from them [61], so boredom proneness level will be higher. Furthermore, Lin and Yu (2008) found that boredom is a crucial motivator for internet addiction [63]. Hence, the relationship between alexithymia and internet novel addiction may be mediated by boredom proneness.

Boredom proneness is likely to occur in college students, so they can seek challenging and stimulating activities to reduce the experience of boredom [64]. Boredom proneness can positively predict the addictive behavior for internet novels. If individuals with a higher degree of alexithymia can reduce the degree of boredom proneness, the degree of addiction to internet novels may also be reduced. If individuals with a high degree of alexithymia continue to pass the role of boredom emotions, the degree of addiction to internet novels will be aggravated to a certain extent. Therefore, college students can reduce the degree of boredom proneness by strengthening activity communication and reduce the effect of alexithymia on internet novel addiction.

### 4.4. Limitations

Several limitations of the present study should be noted. First, the present study is a transversal design. Future studies with a larger number of participants and longitudinal analyses are warranted to elucidate the associations between alexithymia, boredom proneness, and internet novel addiction. Second, the participants in this study were from a single university, so further validation is needed for the generalization of the current findings. Third, the study lacks some demographic information about the participants, such as smoking, weight, alcohol, physical activity, or sedentary status. 

Despite these limitations, this study examined the relationship between alexithymia and internet novel addiction and the mediating role of boredom proneness by constructing a mediation model, which both extends previous relevant research. This study has practical implications for preventing in internet novel addiction among college students. The results of the study suggest that in preventing and intervening in college students’ internet novel addiction, it is important to regulate their boredom proneness level.

## 5. Conclusions

There are positive correlations between college students’ alexithymia, internet novel addiction, and boredom proneness. Alexithymia may directly and indirectly predict internet novel addiction through boredom proneness. This study is expected to promote society’s attention to the behavior of internet novel addiction and guide college students with narrative difficulty to flexibly use boring free time. In future research, more effort could be made to explore the mechanism underpinning the relations among the relevant factors.

## Figures and Tables

**Figure 1 ijerph-19-08708-f001:**
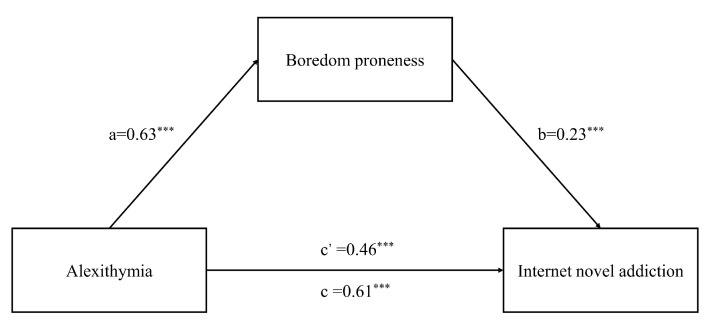
Mediation model between alexithymia, boredom proneness, and internet novel addiction. Note: *** *p* < 0.001.

**Table 1 ijerph-19-08708-t001:** Sex differences in alexithymia, boredom proneness, and internet novel addiction.

	Male	Female	*t*
	*N* = 169	*N* = 175	
**Age** [*M* ± *SD*]	20.64 ± 1.38	20.84 ± 1.31	−1.38
Difficulty identifying feelings [*M* ± *SD*]	18.48 ± 5.49	18.47 ± 5.32	0.04
Difficulty describing feelings [*M* ± *SD*]	14.11 ± 3.00	13.81 ± 3.04	0.94
Externally oriented thinking [*M* ± *SD*]	23.27 ± 2.97	23.25 ± 2.93	0.07
**Total score for alexithymia** [*M* ± *SD*]	55.86 ± 8.03	55.51 ± 8.88	0.38
Internal stimulation [*M* ± *SD*]	35.86 ± 9.16	35.57 ± 8.72	0.30
External stimulation [*M* ± *SD*]	74.74 ± 22.03	74.13 ± 22.23	0.26
**Total score for boredom proneness** [*M* ± *SD*]	110.60 ± 17.57	109.70 ± 18.90	0.65
Health & Learning [*M* ± *SD*]	17.88 ± 5.12	16.47 ± 5.04	2.58 **
Interpersonal relationship [*M* ± *SD*]	12.12 ± 3.59	11.09 ± 3.53	2.68 **
Tolerance [*M* ± *SD*]	18.47 ± 5.28	18.77 ± 5.38	−0.53
Time management [*M* ± *SD*]	12.01 ± 3.56	10.30 ± 3.29	4.63 ***
Withdrawal reaction [*M* ± *SD*]	8.95 ± 2.72	8.44 ± 2.83	1.71
**Total score for internet novel addiction** [*M* ± *SD*]	69.43 ± 18.30	65.07 ± 16.82	2.30 *

Note: * *p* < 0.05, ** *p* < 0.01, *** *p* < 0.001.

**Table 2 ijerph-19-08708-t002:** Means, standard deviation and correlations between all variables (*N* = 344).

Variables	M ± SD	1	2	3
1. Alexithymia	55.69 ± 8.46	1		
2. Boredom proneness	110.14 ± 18.24	0.63 ***	1	
3. Internet novel addiction	67.22 ± 17.67	0.61 ***	0.53 ***	1

Note: *** *p* < 0.001.

**Table 3 ijerph-19-08708-t003:** The mediating role of boredom proneness.

	*p*	Effect	LLCI	ULCI
Mediating effect of boredom proneness	<0.001	0.14	0.16	0.48
Direct effect of alexithymia on internet novel addiction	<0.001	0.46	0.73	1.17
Total effect	<0.001	0.61	1.09	1.44

Note: LLCI, low limit of confidence interval; ULCI, upper limit of confidence interval.

## Data Availability

The data are available from the corresponding authors upon reasonable requests.

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
