# Peer review of "Role of Alexithymia in Predicting Internet Novel Addiction through Boredom Proneness"

_ijerph, 2022, doi:10.3390/ijerph19148708_

Round 1

Reviewer 1 Report

This study has an interesting theme, however needs several adjustments, in order to attain the necessary quality for publication. Details follow:

Introduction:

(+) is well-written, in terms of eloquence and selection of literature. The research question is clearly formulated.

Methods:

(+) this section, though brief, is satisfactorily written;

(+) participants are comprehensively described.

(+) ethical procedures are clearly presented;

(+) statistical analysis is based on appropriate procedures;

Results: (+) comprehensively presented, with a distinct plus for running an additional analysis on gender differences;

Discussion: (+/-)

This section should be further developed, by bringing relevant information about:

-   differences to current literature (i.e. original / new results provided by this study). This refers mostly to sections 4.1 and 4.3

-   inclusion of “Study limitations”. They should include at least the factors influencing the representativity of the findings (e.g., the sampling procedure, running the study in a single university) and their overall strength (e.g., the transversal design).

-   inclusion of a brief section suggesting future research perspectives;

-   practical importance of the study results.

References: (+) are adequately chosen and in concordance with the study theme.

Author Response

This section should be further developed, by bringing relevant information about:

differences to current literature (i.e. original / new results provided by this study). This refers mostly to sections 4.1 and 4.3

Response: We thank you for your suggestion and have added more detail and explanation about the differences to current literature in Sections 4.1 and 4.3. Please see Line 192-199 and Line 252-268.

Sections 4.1: “The results of this study showed that there was significant difference between males and females in terms of Internet novel addiction. Male college students were more addicted to Internet novel than their female counterparts. A study shows that male college students are less focused on health and time management than female college students [43], which might be the internal reasons for male college students to be more addicted to Internet novel. Note that Zhang et al. (2017) did not find the significant correlation be-tween sex and Internet novel addiction [3]. Hence, sex difference should also be considered in the future interventions or preventive strategies to Internet novel addiction. ”

References:

Cluskey, M.; Grobe, D. College Weight Gain and Behavior Transitions: Male and Female Differences. J. Am. Diet. Assoc. 2009, 109, 325-329, doi:10.1016/j.jada.2008.10.045.

Zhang, D.J.; Zhou, Z.K.; Lei, Y.J.; Niu, Q.F.; Zhu, X.W.; Xie, X.C. The Relationship between Neuroticism and Internet Fiction Addiction of College Students: the Mediating Effects of Narrative Transportation and Flow Experience (in Chinese). Psychol. Sci. 2017, 40, 1154-1160.

Sections 4.3: “Most importantly, in the study, boredom proneness was shown to play a mediating role in the impact of alexithymia on Internet novel addiction. The higher the degree of boredom proneness and the degree of alexithymia, the more likely individuals are to develop Internet novel addiction. Alexithymia makes the behavior of Internet novel ad-diction more frequent through the mediating role of boredom proneness. This result is similar to previous findings that alexithymia is positively related to boredom proneness, and problematic behaviors are positively correlated with boredom proneness [57-60]. This result can be explained as follows. Some researchers suggest that boredom can be seen as a psychological conflict whose core feature lies in the lack of emotional awareness, and that people’s inability to directly and deeply experience his or her emotions may be the cause of trait boredom [61]. Individuals with high level of alexithymia are not able to identify, describe, and monitor well their own emotions and those of others, resulting in their internal needs and desires being suppressed [62]. As a result, it is difficult for them to find purpose and meaning in their lives and to gain inspiration and motivation from them [61], so boredom proneness level will be higher. Furthermore, Lin and Yu found that boredom is a crucial motivator for Internet addiction [63]. Hence, the relationship be-tween alexithymia and Internet novel addiction may be mediated by boredom proneness.”

References:

Zhang, Y.; Li, S.; Yu, G. The longitudinal relationship between boredom proneness and mobile phone addiction: Evidence from a cross-lagged model. Curr. Psychol. 2021, doi:10.1007/s12144-020-01333-8.

Wang, W. Exploring the Relationship Among Free-Time Management, Leisure Boredom, and Internet Addiction in Under-graduates in Taiwan. Psychol. Rep. 2019, 122, 1651-1665, doi:10.1177/0033294118789034.

Mercer-Lynn, K.B.; Bar, R.J.; Eastwood, J.D. Causes of boredom: The person, the situation, or both? Pers. Indiv. Differ. 2014, 56, 122-126, doi:10.1016/j.paid.2013.08.034.

Wang, Z.; Yang, X.; Zhang, X. Relationships among boredom proneness, sensation seeking and smartphone addiction among Chinese college students: Mediating roles of pastime, flow experience and self-regulation. Technol. Soc. 2020, 62, 101319, doi:10.1016/j.techsoc.2020.101319.

WANGH, M. Boredom in Psychoanalytic Perspective. Soc. Res. 1975, 42, 538-550.

Kafetsios, K.; Hess, U. Seeing mixed emotions: Alexithymia, emotion perception bias, and quality in dyadic interactions. Pers. Indiv. Differ. 2019, 137, 80-85, doi:10.1016/j.paid.2018.08.014.

Lin, C.; Yu, S. Adolescent Internet usage in Taiwan: Exploring gender differences. Adolesc. 2008, 43, 317.

Li, X.; Feng, X.; Xiao, W.; Zhou, H. Loneliness and Mobile Phone Addiction Among Chinese College Students: The Mediating Roles of Boredom Proneness and Self-Control. Psychol. Res. Behav. Manag. 2021, 14, 687-694, doi:10.2147/PRBM.S315879.

inclusion of “Study limitations”. They should include at least the factors influencing the representativity of the findings (e.g., the sampling procedure, running the study in a single university) and their overall strength (e.g., the transversal design).

inclusion of a brief section suggesting future research perspectives

Response: We thank you for your helpful suggestions. While we summarize the limitations of this study, we also make some suggestions for future research. Please see Line 279-285 in Sections 4.4 and Line 297-299 in Conclusions.

“Several limitations of the present study should be noted. First, the present study is a transversal design. Future studies with a larger number of participants and longitudinal analyses are warranted in order to elucidate the associations between alexithymia, boredom proneness, and Internet novel addiction. Second, the participants in this study were from a single university, so further validation is needed for the generalization of the current findings. Third, the study lacks some demographic information about the participants, such as smoking, weight, alcohol, physical activity, or sedentary status.”

“In future research, more effort could be made to explore the mechanism underpinning the relations among the relevant factors.”

 practical importance of the study results.

Response: We thank you for your valuable comments. We illustrate some of the practical implications of the study from the results of this study. We have added this part in Line 286-291.

“Despite these limitations, this study examined the relationship between alexithymia and Internet novel addiction and the mediating role of boredom proneness by con-structing a mediation model, which both extends previous relevant research. This study has practical implications for preventing in Internet novel addiction among college students. The results of the study suggest that in preventing and intervening in college students' Internet novel addiction, it is important to regulate their boredom proneness level.”

Reviewer 2 Report

In this article, the authors investigated the relationship between alexithymia, boredom and Internet addiction to novels. The results showed that there was no significant difference between female and male college students in terms of alexithymia and boredom propensity, while male college students had a higher total Internet novel addiction score than females. There were significant positive correlations between alexithymia, boredom tendency, and new Internet addiction. The propensity to boredom has played a partial mediating role in the impact of alexithymia on the new Internet addiction. The statistical analysis was performed correctly and with correct methods. In particular, the authors evaluated Harman's single factor test for all items. It showed that the relationship between alexithymia, boredom and Internet addiction to novels was explain by other factors. Despite this, in this article the authors do not report the descriptive statistical table. The descriptive statistical table would be very important to evaluate several variables: age (male and female), smoking, weight, alcohol, physical activity or sedentary state which are fundamental especially when the authors decide to compare two groups. So, I think that would be necessary to implement the statistical descriptive table. Secondly, in the "Materials and Methods" section the authors specified that: “There were 169 males (49.13%) and 85 175 females (50.87%): 67 first-year students (19.48%), 80 second-year students (23.26%), 88 86 third-year students (25.58%), and 109 fourth-year students (31.69%)” but the Table 1 show the difference between total male and female. If it has not been considered before, authors must use the correct statistical method to stratify the two groups by student-year because it could be a selection bias.

Author Response

Despite this, in this article the authors do not report the descriptive statistical table. The descriptive statistical table would be very important to evaluate several variables: age (male and female), smoking, weight, alcohol, physical activity or sedentary state which are fundamental especially when the authors decide to compare two groups. So, I think that would be necessary to implement the statistical descriptive table.

Response: We thank you for your valuable comments. But we are very sorry for not collecting these important variables (smoking, weight, alcohol, physical activity or sedentary status) in addition to age and gender when doing the study. Thus, we mentioned this in the limitations of the study. Please see Line 284-285 in Sections 4.4.

“Third, the study lacks some demographic information about the participants, such as smoking, weight, alcohol, physical activity, or sedentary status.”

Secondly, in the "Materials and Methods" section the authors specified that: “There were 169 males (49.13%) and 85 175 females (50.87%): 67 first-year students (19.48%), 80 second-year students (23.26%), 88 86 third-year students (25.58%), and 109 fourth-year students (31.69%)” but the Table 1 show the difference between total male and female. If it has not been considered before, authors must use the correct statistical method to stratify the two groups by student-year because it could be a selection bias.

Response: We appreciate your suggestion. Before sending the questionnaires, we had information about the ratio of male to female in each grade in the surveyed university. The current test situation is more in line with expectations. Please see Line 94-96.

“The male-female ratio in all grades was approximately in line with the real situation of the university surveyed.”

Reviewer 3 Report

I Thank the editor for the possibility to review this interesting manuscript. I appreciated the authors’ clearness in developing and testing their hypotheses.  Also I believe the results to be of interest for a generalist public and to focus on a very hot topic for what concerns mental health. I also appreciate the attempt to explore possible “affective” factors influencint internet addiction. Generally speaking I think the manuscript is very well structured and I just have some minor questions. Please, find below some minor points which I believe can significantly improve the paper readability and results’ dissemination.

How was the sample size calculated? Did authors performed a power analysis? Please provide more information.

On the mediation model, why did the authors hypothesized that boredom mediates between alexithymia and internet addiction and not the reverse (X = boredom, M =  alexithymia)? Please, specify more clearly why you choose this direction.

Please, better discuss the possible role of boredom in influencing the perceived time during internet usage. For example, consider for the discussion’s integration recent evidence suggesting that cognitive absorption, in the form of temporal dissociation during usage, predicts social network addiction .  A similar relationship was also discussed in relationship to gaming addiction .  If we consider boredom a sort of oppost state as compared to cognitive absorption (boredom = no engagement and cognitive absorption = high engagement) how is it possible that both of them are expected to positibely predict addiction? Based on this previous literature, I suggest to the author to better define an operational difference between boredom in general in one’s own life (as I think the author measured) and boredom during the usage as a consequence of the usage itself.

Author Response

How was the sample size calculated? Did authors performed a power analysis? Please provide more information.

Response: We thank you for your concern. We have added more detail about the sample size. Please see Lines 88-90 in Participants.

“The determination of sample size was conducted using WebPower [38]. With the path coefficients of 0.3, power of 0.95, and 3 predictors, the smallest sample size required was 290.”

References:

Zhang, Z.; Mai, Y.; Yang, M.; Zhang, M.Z. Package 'WebPower'. Available online: http://mirror.nju.edu.cn/CRAN/web/packages/WebPower/WebPower.pdf (accessed on 20 March 2022).

On the mediation model, why did the authors hypothesized that boredom mediates between alexithymia and internet addiction and not the reverse (X = boredom, M =  alexithymia)? Please, specify more clearly why you choose this direction.

Response: We thank you for your suggestions. We have added more detail and explanation about this direction. Please see Line 74-85.

“Moreover, Eastwood et al. suggests that the lack of emotional awareness is the main reason for the experience of boredom [34]. Therefore, alexithymia may be one of the fac-tors affecting the experience of boredom due to the inability to better experience and ex-press their emotions [34]. Extant literature has found that alexithymia has a significant predictive effect on boredom proneness, alexithymia and Internet addiction are interrelated, and boredom proneness and Internet addiction are significantly positively correlated. However, whether Internet novel addiction as a sub-category of Internet addiction is related to alexithymia and boredom proneness requires further study. Therefore, to explore the relationship between college students' alexithymia, Internet novel addiction, and boredom proneness, 344 college students were recruited for the present study. We hypothesized that boredom proneness may play a mediating role in the prediction of alexithymia to Internet novel addiction.”

References:

Eastwood, J.D.; Cavaliere, C.; Fahlman, S.A.; Eastwood, A.E. A desire for desires: Boredom and its relation to alexithymia. Pers. Indiv. Differ. 2007, 42, 1035-1045, doi:10.1016/j.paid.2006.08.027.

Please, better discuss the possible role of boredom in influencing the perceived time during internet usage. For example, consider for the discussion’s integration recent evidence suggesting that cognitive absorption, in the form of temporal dissociation during usage, predicts social network addiction.  A similar relationship was also discussed in relationship to gaming addiction.  If we consider boredom a sort of oppost state as compared to cognitive absorption (boredom = no engagement and cognitive absorption = high engagement) how is it possible that both of them are expected to positibely predict addiction? Based on this previous literature, I suggest to the author to better define an operational difference between boredom in general in one’s own life (as I think the author measured) and boredom during the usage as a consequence of the usage itself.

Response: We thank you for your valuable comments. First, in this study we specify the definition of boredom proneness, second, we discuss the possible role of boredom in influencing perceived time during Internet usage. Please see Lines 65-67 and 235-250.

“In this study, the boredom proneness refers to stable individual differences in boredom emotional responses and behaviors among relatively persistent personality traits.”

“On the other hand, Vodanovich and Kass (1990), extracted five orthogonal factors from the BPS, including External Stimulation, Internal Stimulation, Constraints, Affective Response, and Perception of Time [53]. In the present study, the BPS only have two dimensions: external stimulation and internal stimulation. Huang et al. (2010) attributes time perception to self-control in internal stimulation, that is, high scorers tend to lose their attention, and it is difficult to maintain long-term attention to a single person, object or thing [22]. Therefore, according to the attention theory of boredom proneness [54], the root of an individual’ s susceptibility to boredom lies in the interruption of attention regulation; that is, the inability to maintain and regulate attention [35,55]. The present results also fit the Person-Affect-Cognition-Execution model [56], in which boredom proneness affects individuals' cognitive and emotional responses, and enhances their de-sire to reduce their boredom experience by using a certain application. Specifically, individuals with high levels of boredom proneness have difficulty directing their attention to a stimulus or trying other things to alleviate boredom, and they tend to immerse them-selves in Internet novels to escape from real problems and boredom, and therefore tend to develop Internet novel addiction.”

References:

Vodanovich, S.J.; Kass, S.J. A Factor Analytic Study of the Boredom Proneness Scale. J. Pers. Assess. 1990, 55, 115-123, doi:10.1080/00223891.1990.9674051.

Huang, S.H.; Li, D.L.; Zhang, W.; Li, D.P.; Zhong, H.R.; Huang, C.K. The Development of Boredom Proneness Ques-tionnaire for College Students (in Chinese). Psychological Development and Education. 2010, 26, 308-314.

Harris, M.B. Correlates and Characteristics of Boredom Proneness and Boredom. J. Appl. Soc. Psychol. 2000, 30, 576-598, doi:https://doi.org/10.1111/j.1559-1816.2000.tb02497.x.

Eastwood, J.D.; Frischen, A.; Fenske, M.J.; Smilek, D. The Unengaged Mind: Defining Boredom in Terms of Attention. Perspect Psychol Sci 2012, 7, 482-495, doi:10.1177/1745691612456044.

Hunter, A.; Eastwood, J.D. Does state boredom cause failures of attention? Examining the relations between trait boredom, state boredom, and sustained attention. Exp. Brain Res. 2018, 236, 2483-2492, doi:10.1007/s00221-016-4749-7.

Brand, M.; Young, K.S.; Laier, C.; Wölfling, K.; Potenza, M.N. Integrating psychological and neurobiological considerations regarding the development and maintenance of specific Internet-use disorders: An Interaction of Per-son-Affect-Cognition-Execution (I-PACE) model. Neuroscience & Biobehavioral Reviews 2016, 71, 252-266, doi:10.1016/j.neubiorev.2016.08.033.

Round 2

Reviewer 1 Report

Compared to the first version, the second version of the paper seems clearer and addresses the flaws of the previously submitted manuscript. It could be published in its current form.